# Effect of Cathodic Protection Potential Fluctuations on the Corrosion of Low-Carbon Steels and Hydrogen Absorption by the Metal in Chloride Solutions with Nearly Neutral pH

**DOI:** 10.3390/ma15238279

**Published:** 2022-11-22

**Authors:** Andrey I. Marshakov, Alevtina A. Rybkina

**Affiliations:** Frumkin Institute of Physical Chemistry and Electrochemistry, Russian Academy of Sciences, Leninsky Prospect 31-4, Moscow 119071, Russia

**Keywords:** low-carbon steel, sign-alternating polarization, weight loss, pitting corrosion, hydrogen absorption

## Abstract

Considerable fluctuations in the cathodic protection potential under the impact of stray currents lead to the occurrence of local corrosion on steel structures operated in soils and seawater. The potential fluctuations induced by both alternating and direct current sources can be simulated by cycling a square potential step. This paper covers the impact of sign-alternating cyclic polarization (SACP) on the general and local corrosion of carbon steel in 3.5% NaCl solution containing a borate buffer (pH 6.7) and without it. A decrease in the cathodic half-period potential (*E*_c_) of SACP inhibited the general corrosion and accelerates the local corrosion of steel in both solutions, which was associated with an increase in the amount of hydrogen in the metal. Increasing the duration of the SACP cathodic half-period increased the pit density and total area at less negative *E*_c_ values. At more negative *E*_c_ values, an increase in the duration of cathodic polarization reduced the intensity of steel local corrosion in the unbuffered chloride solution. This effect is explained by blocking of the pit nucleation centers on the metal surface by a layer of steel dissolution products formed in the near-electrode electrolyte layer with a high pH. The combined body of data shows that hydrogen absorption by the metal determines the corrosion behavior of carbon steel under the impact of SACP in chloride solutions, which should be taken into account in the development of models of the corrosion of steel structures under the action of stray currents.

## 1. Introduction

Metal corrosion caused by stray currents is one of the main reasons for the destruction of steel structures operated in soils or in seawater. Cathodic protection of structures is not always capable of preventing the development of this type of corrosion. Significant fluctuations in the cathodic protection potential under the action of stray currents lead to the occurrence of steel corrosion of local pitting [1,2,3,4,5] and, possibly, stress corrosion cracking [6,7,8,9,10]. In the presence of stray currents, improving the quality of an insulation coating does not guarantee the safety of metal structures because a decrease in the number and size of through defects in the coating results in an increase in the density of current through the metal/electrolyte interface, and thus can accelerate the growth of a corrosion defect. Thus, the creation of predictive models of the corrosion of steel structures under the impact of stray currents is of great importance. To this end, a mechanistic relationship between potential fluctuations and pit initiation needs to be ascertained.

As a rule, a distinction is made between the action of stray currents from DC sources (DC corrosion) and currents induced by AC sources, primarily high-voltage power lines (AC corrosion) [11]. At the same time, the magnitude and polarity of stray currents from DC sources may vary with time which, given the structure is under cathodic protection, leads to potential fluctuations, and its instantaneous value can become either more positive or more negative than the free corrosion potential of steel [12,13]. In view of this, the potential fluctuations with time induced by both alternating and direct current sources were simulated by square potential pulses. This method of simulating stray currents has been used in various versions and, accordingly, has been given various names such as the anodic transient test after cathodic protection [14,15,16], square wave polarization (SWP) [17,18,19,20,21,22], and sign-alternating cycling polarization (SACP) [23]. SACP is a version of SWP where the potential jumps between its cathodic and anodic values.

Simulating stray currents by square potential pulses makes it possible to take non-faradic capacitive currents, whose share in the case of a sinusoidal current (or voltage) variation in the total current can be significant. Hence, it is possible to obtain an unambiguous relationship between the potential pulse amplitude and the algebraic sum of the rates of faradic processes occurring on the electrode. It is also necessary to take into account the quasi-capacitive currents associated with metal oxidation during the anodic potential pulse and reduction of the oxide during the cathodic pulse. However, quasi-capacitive currents also appear in the case of a sinusoidal variation in the electrode potential [24].

When the potential pulses are square, as under the action of an alternating current, the formation of pits is observed [14,15,16,17,18,19,20,21,22,23]. The local nature of steel corrosion can be explained by a significant increase in the pH of the near-electrode layer of the solution when the cathodic current is flowing. On applying an anodic potential in an alkaline environment, passivation of the metal can occur. If the amplitude of the potential jump and the duration of the anodic transient exceed critical values, breakdown of the passive film occurs and the steel dissolves locally [14,15,16,17]. This concept is consistent with the mechanism of AC corrosion of steel pipes in soil [1,11]. However, pits are formed on the surface of steel pipe when the potential is cycled between values significantly lower than the pitting potential [18,19,20,21,22,23], and pits are formed in a mixture of a simulated soil electrolyte with a buffer solution [24]. These effects show that the explanation of pit formation on a passive steel surface during the SWP anodic half-period is not universal.

The local dissolution of pipe steel in the simulated soil electrolyte under SWP impact has been explained by a shift in the potential of a double electric layer in the positive direction [21,22,23]. It was shown that the pit density increases with an increase in frequency [22] and amplitude [23] of SWP. The majority of corrosion pits are generated in the steel matrix rather than on inclusions, since when the potential fluctuates, the steel matrix is the anode, while the cathodic reaction can occur on non-metallic inclusions [21,22,23]. It should be noted that this concept does not take into account the possibility of pit formation due to the absorption of hydrogen atoms and the formation of high-pressure points in the steel matrix [25]. It has been shown that the pit density increases with an increase in the duration of the cathodic half-cycle with in a cyclic potential pulse, and on addition of a hydrogen absorption promoter (thiourea) to the electrolyte [24]. The anodic current that appears after switching the potential from a cathodic to an anodic value can be determined by the rate of extraction and ionization of hydrogen from iron [26], and the intensity of steel local corrosion correlates with the anodic current value under SWP [24].

At the same time, hydrogen absorbed by the metal can inhibit the anodic dissolution of iron [27,28]. Apparently, hydrogen atoms passing from the absorbed to adsorbed state block the metal dissolution centers, such as oxygen atoms, halide ions, and species of organic inhibitors. Hence, adsorbed hydrogen can have a dual effect on the dissolution of iron and steel, and can promote the formation of pits but inhibit the active dissolution of a metal.

Thus, the corrosion rate of steels under SWP may depend on a number of effects: (i) an increase in the near-electrode pH and passivation of the electrode, (ii) a non-equilibrium state of the double electric layer, and (iii) the amount of hydrogen absorbed by the metal. It is of undoubted interest to determine which of these effects is predominant in different modes of metal polarization and in solutions with various compositions.

The purpose of this work is to study the effect of alternating polarization on the general and local corrosion of low-carbon steel and on the rate of hydrogen absorption by the metal in both buffered and unbuffered chloride solutions with nearly neutral pH values. Varying the cathodic potential and the duration of the SACP cathodic half-period with measurement of the rate of hydrogen penetration into the metal should show the role of absorbed hydrogen in the corrosion of steel under the impact of stray currents. The difference in the corrosion behavior of steel in buffered and unbuffered solutions should show the effect of the near-electrode pH on the local corrosion of steel under SACP.

## 2. Materials and Methods

Samples made from a thin sheet of 08 kp grade rolled steel were used in the study. The chemical composition of the steel is shown in Table 1. Samples size was 20 × 25 × 0.1 mm. A foil strip of the same steel grade was used as the current supply line to the electrode. Electrical contact was made by connecting the electrode and the current supply line in a clamp made of an inert material, which was totally immersed in the solution to avoid corrosion losses of the electrode at the solution-air interface. The electrode surface was abraded with SiC paper up to 600 grit. The electrodes were then washed in a Sapphire–0.8 TT ultrasonic bath in a C_2_H_5_OH:C_7_H_8_OH 1:1 mixture for 10 min, washed with deionized water, and dried. After immersion in the electrolyte solution, the electrode was cathodically polarized at a potential of −0.65 V (SHE) for 10 min to remove the air-formed oxide film from the surface.

The test solutions included a 3.5% aqueous NaCl solution (NaCl solution) and a solution with the same NaCl concentration based on borate buffer with pH 6.7 (NaCl + BB solution). The composition of the borate buffer was 0.4 M H_3_BO_3_ + 5.5 mM Na_2_B_4_O_7_ × 10 H_2_O. The polarization curves obtained on iron and mild steel in this buffer solution have been reported previously [26,29].

All the solutions were prepared from reagents of “chemically pure” grade and distilled water. The tests were carried out at ambient temperature (20 ± 2 °C).

The electrode was polarized in a three-electrode cell with separated anodic and cathodic spaces using an IPC-Pro MF potentiostat (“Volta”, Saint Petersburg, Russia). A platinum auxiliary electrode and a silver chloride reference electrode were used. The potentials are reported versus SHE. The potential setting time to within 1 mV was no more than 5 × 10^−4^ s.

The potential of the SACP anodic half-period (*E*_a_) was always −0.3 V, and the potential of the cathodic half-period (*E*_c_) was −0.60, −0.80, −1.05, or −1.15 V. The duration of the SACP cathodic half-period (τ_c_) was 10, 33, 100, or 1000 ms. The duration of the anodic half-period (τ_a_) was 10 ms. The number of SACP cycles in all experiments was the same (10^6^ cycles), and hence the total duration of anodic polarization of the electrode was 10^4^ s. The results obtained upon SACP were compared with those obtained upon potentiostatic (−0.3 V) polarization of the electrode whose duration was also 10^4^ s. The free corrosion potential of steel (*E*_corr_) in the NaCl and NaCl+BB solutions was −0.46 ± 0.01 V and −0.42 ± 0.01 V, respectively. As schematic diagram of the potential variation under SACP is shown in Figure 1.

The weight loss of samples upon SACP (*K*_imp_) and upon potentiostatic polarization of the sample (*K*_st_) was determined by a gravimetric method using an AF-R220CE analytical balance (manufactured by SHINKODENSHI Co. Ltd. (Vibra) Itabashi, Japan). Corrosion products were removed from the sample after testing using a HCl solution (1:1) with addition of urotropin (0.5% C_6_H_12_N_4_), then the sample was washed in distilled water and dried with filter paper. The weight loss of samples per unit area was calculated by Equation (1):(1)$$K_{\mathrm{imp}}(K_{\mathrm{st}})\;=\;\frac{m_{0}-m_{1}}{S},$$
where *m*_0_ is the weight of the sample before testing, *m*_1_ is the sample weight after removal of corrosion products, and *S* is the sample surface area.

After testing, the sample surface was photographed using a Biomed PR-3 optical microscope (manufactured by Biomed Service LLC, Moscow, Russia) and an AC-300 digital video camera (Amoyca, Ho Chi Minh City, Vietnam) connected to its eyepiece. The camera resolution was 2048 × 1536 pixels. Data from the camera were evaluated with Scope Photo 3.0 software (“Scope Tec”, Munich, Germany) to determine the number of pits (*N*) and the surface area of the sample occupied by each pit (*S*_i_). The *S*_i_ values were determined as the number of pixels in the contour of a selected defect and were then converted to area units (mm^2^). Initially, *N* and *S*_i_ were determined from surface images at a low magnification and refined using images with a higher magnification (Figure 2).

The density of sample surface coverage with defects (ρ) was determined using Equation (2):ρ = *N*/*S*(2)
and the total surface area of the sample (*S*_p_) occupied by pits was determined using Equation (3)
(3)$$S_{p}\;=\;\sum_{i\;=\;1}^{N}S_{i}/S_{0}$$
where *S*_0_ is the visible sample area (1.00 mm^2^).

The difference in the values of *S*_p_ and ρ obtained on different parts (1 mm^2^) of the working surface area did not exceed 15% of the mean values.

The pit diameter (*d*_i_) was determined with the assumption that the pit has a hemispherical shape. The number of pits with *d*_i_ smaller than 5 µm, from 5 to 10 µm, and larger than 10 µm was determined. The *d*_i_ of pits on the electrode surface after SACP usually did not exceed 20 µm.

The measurement of the hydrogen penetration current through the steel membrane under SACP and under cathodic polarization was performed in a cell developed by Devanathan and Stahurski [30]. The membrane was 3.5 cm in diameter and 0.1 mm thick, and the surface area in contact with the electrolyte was 3 cm^2^. A palladium layer was applied on the diffusion side of the membrane, as described in [31]. The diffusing part of the cell was filled with 0.1 M NaOH, and the membrane was polarized at a potential of 0.45 V (SHE). The flow of hydrogen through the membrane was measured as its faradic equivalent *i*_p_ = *i* − *i*_bg_, where *i*_bg_ is the background current density. The background current density was less than 5 × 10^−3^ A/m^2^. The working side of the membranes was prepared in the same way as the surface of the samples during the above experiments. The duration of the experiments under SACP was 600 s.

## 3. Results and Discussion

Figure 3 shows the effect of the cathodic half-period potential (*E*_c_) of SACP on the steel weight loss (*K*_imp_) in NaCl (curve 1) and NaCl + BB (curve 2) solutions. The steel weight loss at a constant potential *E* = −0.3 V (*K*_st_) is shown in Figure 3 by dotted and dot-and-dashed lines in NaCl and NaCl + BB solutions, respectively. The *K*_st_ value was determined for a time of 10^4^ s, which is equal to the total duration of anodic polarization of steel under SACP. As can be seen, a significant inhibition of steel dissolution occurred in both solutions with a decrease in the *E*_c_ value, but the degree of inhibition was different. At *E*_c_ = −1.05, the *K*_st_/*K*_imp_ ratios of NaCl and NaCl + BB solutions were 30 and 3.6, respectively.

The effect of reducing the iron dissolution rate under SACP imposition was observed in pure borate buffer (pH 7) and was explained by the effect of adsorbed hydrogen [26]. It was previously shown that an increase in the degree of hydrogen absorption on the iron surface inhibits its anodic dissolution at a constant potential in acid electrolytes [32,33] or at the initial stage of its dissolution in a neutral electrolyte [27].

A decrease in *E*_c_ should accelerate the discharge of H^+^ ions and, accordingly, increase the degree of hydrogen absorption of the metal surface. An increase in the amount of adsorbed hydrogen on iron or steel can be inferred from the rate of hydrogen penetration through the metal membrane, *i*_p_ [34,35]. Figure 4 shows the dependence of *i*_p_ on *E*_c_ under SACP in the NaCl (curve 1) and NaCl + BB (curve 2) solutions. In the NaCl solution, *i*_p_ increased with a decrease in *E*_c_ to a greater extent than in the NaCl + BB solution, which was qualitatively consistent with a sharper decrease in *K*_imp_ with a decrease in *E*_c_.

It should be noted that in the NaCl + BB solution, the *K*_imp_ value at *E*_c_ = −0.6 V slightly exceeded *K*_st_ (Figure 3). Apparently, this is due to the fact that in a neutral borate solution, at potentials of the so-called active iron dissolution, a primary passive film exists on its surface [28,36,37,38]. It was shown by electrochemical nano-weighing on an EQCN quartz crystal that at *E* = −0.3 V, the mean thickness of this film reached several monolayers of oxide/hydroxide compounds [28]. Atomic hydrogen reduces these compounds, thus accelerating the dissolution of iron or carbon steel in neutral electrolytes [25]. Therefore, a superposition of two effects should be observed in the NaCl + BB solution under the impact of SACP: acceleration of metal dissolution due to oxide film reduction and inhibition of its dissolution due to adsorption of H atoms on the pure metal surface. As a result, *K*_imp_ was larger than *K*_st_ at less negative *E*_c_ values, but *K_i_*_mp_ became smaller than *K*_st_ as *E*_c_ decreased.

The impact of SACP on the intensity of local steel corrosion, i.e., on the density of pits (ρ) and their total area (*S*), was studied along with steel weight loss determination. Figure 5A shows the variation in pit density with a decrease in *E*_c_ (the ρ values under conditions of steel dissolution at a constant potential are shown in the figure at *E*_c_ = *E*_a_ = −0.3 V). In NaCl solution, the ρ value increased from 12 pits/mm^2^ at a constant potential of −0.3 V to 195 pits/mm^2^ under SACP with *E*_c_= −1.05 V (curve 1). In the NaCl + BB solution, the ρ value increased to a slightly greater extent from 9 to 240 pits/mm^2^ at *E*_a_ = −0.3 V and *E*_c_ = −1.05 V, respectively (curve 2). An increase in the pit density in the buffer solution showed that an increase in the near-electrode pH under SACP was not the main reason for pit formation on steel.

The total area of pits also increased with a decrease in *E*_c_ in both solutions (Figure 5B). Moreover, the increase in *S* was associated not only with an increase in ρ but also with an increase in the pit sizes. Figure 6 shows the change in the number of pits up to 5 µm, from 5 to 10 µm, and larger than 10 µm in diameter, depending on *E*_c_ value. As one can see, in the NaCl solution with more negative *E*_c_ values (−0.8 and −1.05 V), pits larger than 10 µm in diameter appeared on the steel surface (Figure 6A). At the same *E*_c_ values, pits with smaller diameters appear on the steel in the NaCl + BB solution (Figure 6B).

The data obtained (Figure 3, Figure 4, Figure 5 and Figure 6) are consistent with the concept of the diverse effects of atomic hydrogen on the iron anodic dissolution rates and on steel local corrosion. The H atoms adsorbed on the metal surface inhibited its uniform dissolution at a constant potential [26,27,28]. The H atoms absorbed by the metal favored the generation of pits [24,25]. Since exchange of hydrogen atoms between the metal’s surface and phase occurs [34,35], these two effects can be observed simultaneously with an increase in the hydrogen concentration in the metal.

At the same time, the rate of hydrogen incorporation into steel is not the only factor that affects the intensity of local corrosion under SACP. The pit growth is accelerated by anions as activators of the anodic dissolution of iron, in particular by Cl^-^ ions [24]. The borate ion inhibits the dissolution of iron [24] and can prevent the formation of larger pits (Figure 6B).

Despite some differences in the corrosion behavior of steel under SACP in buffered and unbuffered solutions, the main effects were the same in both solutions, i.e., the general corrosion of steel decreased under SACP, whereas the intensity of local corrosion increased. Therefore, an increase in the pH of the near-electrode layer was not the main cause of steel corrosion under SACP. In view of this, further studies were continued only in the NaCl solution.

An increase in the duration of the SACP cathodic half-period (τ_c_) should lead to the accumulation of absorbed hydrogen in steel and, accordingly, to an increase in both the degree of hydrogen coverage of the electrode surface and the concentration of H atoms in the metal phase. As a result, the total steel dissolution rate should decrease. In fact, in the NaCl solution under SACP (−0.3 ↔ −0.6 V), an increase in τ_c_ from 10 ms to 1 s resulted in a 1.6-fold decrease in steel weight loss of 1.99 and 0.43 mg/cm^2^, respectively. Under SACP in the (−0.3 ↔ −0.8 V) mode, an increase in τ_c_ from 10 ms to 33 ms reduced the steel dissolution rate from 1.19 to 0.22 mg/cm^2^. It should be noted that the total duration of the SACP anodic half-period remained constant in all the experiments.

An increase in τ_c_ at *E*_c_ = −0.6 and −0.8 V favored a more intense local steel corrosion (Figure 7, curves 1 and 2). The density of pits (Figure 7A) and their total area (Figure 7B) under conditions of steel dissolution at a constant potential are shown in the figures at τ_c_ = 0. It can be seen that under SACP in the (−0.3 ↔ −0.6 V) mode the pit density increased with an increase in τ_c_ to 33 ms and then remained constant (Figure 7A, curve 1). Their total area continues to grow with an increase in the duration of the cathodic half-period (Figure 7B, curve 1), which was associated with the formation of larger pits (Figure 8A). Under SACP in the (−0.3 ↔ −0.8 V) mode, the highest ρ value was observed at τ_c_ = 10 ms (Figure 7A, curve 2), but total area of pits increased at τ_c_ = 33 ms (Figure 7A, curve 2) since larger pits appeared.

Under SACP in the (−0.3 ↔ −1.05 V) mode, an increase in the duration of the cathodic half-period has a qualitatively different effect on the intensity of local corrosion than at less negative *E*_c_ values. With an increase in τ_c_ from 10 ms to 1 s, both the pit density (Figure 7A, curve 3) and the total area of pits decrease (Figure 7B, curve 3), while large defects are not generated (Figure 8B).

To understand the possible reasons for the different effects of the SACP cathodic half-period duration at different *E*_c_ values on the intensity of local steel corrosion, we adjusted the following ratio:*k* = *i*_p_/*i*_p.st_,(4)
where *i*_p_ and *i*_p.st_ are the rates of hydrogen penetration into steel under SACP and under potentiostatic conditions, respectively, at a potential of *E*_c_.

Since there are almost no corrosion defects on the steel surface at a constant cathodic potential, *i*_p.st_ corresponds to the hydrogen absorption on the entire electrode surface. At the same time, there may be places of preferential hydrogen absorption associated with grain boundaries, inclusions, dislocations, and other crystal lattice defects. If at the SACP anodic potential the rate of penetration into the metal is negligible and anodic polarization of the electrode does not affect the conditions of hydrogen absorption during the cathodic period, then
*i*_p_ = *i*_p.st_  τ_c_/(τ_c_ + τ_a_),(5)
and from (4) and (5) we obtain:*k*^theor^ = τ_c_/(τ_c_ + τ_a_),(6)

The plot of *k*^theor^ vs. τ_c_ calculated by Equation (6), on the assumption that anodic polarization does not affect the hydrogen absorption rate, is shown in Figure 8 by the dashed line. However, it has been repeatedly shown that the cathodic release and incorporation of hydrogen into iron or steel can occur under anodic polarization at higher rates than it would follow from the extrapolation of the Tafel sections of polarization curves obtained at cathodic potentials (see e.g., [39,40]). In this case, the *i*_p_ value would be larger than that calculated by Equation (5), and hence, the experimental *k* value would be larger than that predicted by Equation (6). Indeed, under SACP (−0.3 ↔ −0.6 V), the *k* factor for all τ_c_ values was not only larger than the *k*^theor^ value but even exceeded 1.0 (Figure 9, curve 1). A possible reason for this effect is the local acidification of the electrolyte inside a pit, which should lead to acceleration of the processes of hydrogen cathodic evolution and absorption both during the anodic and cathodic SACP periods.

Under SACP in the (−0.3 ↔ −0.8 V) mode, the *k-*τ_c_ curve (Figure 9, curve 2) was close to that calculated by Equation (6) (dashed line), but if τ_c_ = 10 ms, the experimental *k* value exceeded *k*^theor^, and with an increase in the duration of the cathodic half-period, for example, at τ_c_ = 1 s, the opposite effect was observed. Under SACP in the (−0.3 ↔ −1.05 V) mode, *k* was significantly smaller than *k*^theor^ at all τ_c_ values (Figure 9, curve 3), that is, the SACP value was smaller than that observed under potentiostatic polarization of the electrode.

It is obvious that under conditions of electrode cyclic polarization, the iron ions formed during the anodic period subsequently pass from the internal pit space into the electrolyte near-electrode layer with a high pH outside the pit and form a layer of oxide/hydroxide products of steel dissolution around the corrosion defect. The formation of a layer of insoluble corrosion products on a steel surface under alternating polarization has been observed in many studies, for example, in [15,16]. Apparently, this layer blocks the hydrogen absorption centers on the metal surface, resulting in *k* < *k*^theor^. Iron oxide/hydroxide compounds can be reduced at sufficiently negative potentials, but this process cannot occur instantaneously. Indeed, under SACP in the (−0.3 ↔ −1.15 V) mode, *k* increased with an increase in the duration of the cathodic half-cycle, and at τ_c_ = 1 s it approached the calculated value of *k*^theor^ (Figure 9, curve 4).

It can be assumed that the layer of steel dissolution products blocks the most active hydrogen absorption sites and prevents the formation of pits on these places on the metal surface. Then, the reduction of iron oxide/hydroxide compounds should favor more intense local corrosion. Indeed, at the same duration (100 ms) of the SACP cathodic half-period at *E*_c_ = −1.15 V, the pit density and total pit area increased compared to the values observed at *E*_c_ = −1.05 V (Figure 7, point 4 and curve 3).

Thus, an increase in the duration of the SACP cathodic half-period can both accelerate and inhibit the local corrosion of steel in a chloride solution. At less negative cathodic potentials (−0.6 and −0.8 V), the increase in τ_c_ speeds up pitting corrosion, whereas at a more negative value (−1.05 V) it slows down the process. We believe that the latter effect is associated with the formation of a layer of insoluble iron compounds on the steel surface, which blocks the centers of hydrogen absorption and pit nucleation. The reason for the formation of this layer lies in a pH increase in the near-electrode solution layer, while an increase in the SACP cathodic half-period duration should lead to an increase in the layer thickness or surface area of the electrode occupied by corrosion products. When the cathodic half-period potential is −1.15 V, reduction of insoluble iron compounds apparently occurs and the intensity of local corrosion increases.

## 4. Conclusions

(1)With a decrease in the SACP cathodic half-period potential (*E*_c_), the rate of general corrosion of carbon steel determined by the gravimetric method decreased both in buffered and unbuffered 3.5% NaCl solutions. A decrease in the rate of general corrosion under SACP correlated with an increase in the amount of absorbed hydrogen determined from the rate of hydrogen penetration through a steel membrane, which is consistent with the effect of anodic dissolution inhibition of iron at a constant potential and an increase in the degree of hydrogen coverage of the electrode surface [26,27,28,32,33].(2)With a decrease in *E*_c_, the intensity of steel local corrosion, namely, the pit density and the total area of pits, increased both in buffered and unbuffered chloride solutions. Hence, an increase in the near-electrode solution pH is not the main cause of steel pitting corrosion under SACP.(3)An increase in the cathodic half-period duration increased the pit density and total area at less negative *E*_c_ values (−0.6 and −0.8 V), which correlated with an increase in the amount of hydrogen absorbed by the metal. At *E*_c_ = −1.05 V, the opposite effect was observed, which might be explained by deposition of iron oxide/hydroxide compounds on the metal. These compounds are insoluble in the near-surface electrolyte layer with a high pH. As a result, the layer of steel dissolution products blocks the centers of pit nucleation.(4)Based on the combination of data obtained, we conclude that the absorption of hydrogen by the metal determines the corrosive behavior of carbon steel under SACP in chloride solutions, i.e., it reduces the rate of general corrosion and increases the intensity of pitting corrosion.

## Figures and Tables

**Figure 1 materials-15-08279-f001:**
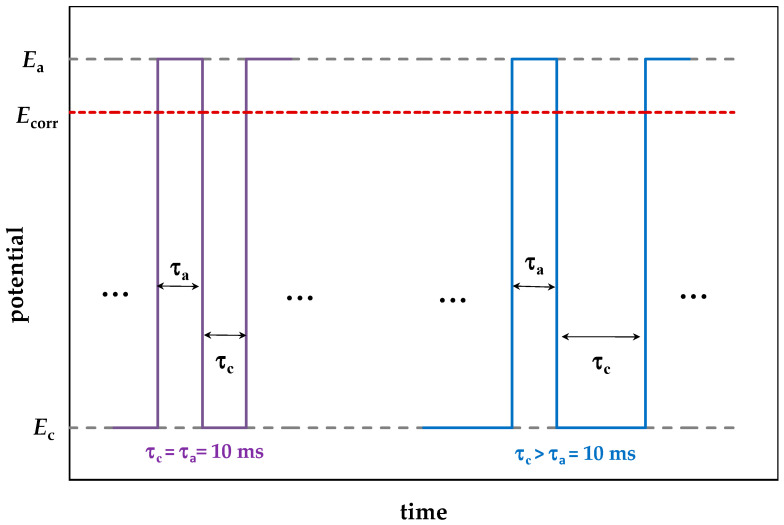
Schematic diagram of potential under sign-alternating cyclic polarization, where *E*_a_ and *E*_c_ are the potential values during the anodic and cathodic half-periods, *E*_corr_ is the free corrosion potential of steel, and τ_a_ and τ_c_ are the durations of the anodic and cathodic half-periods.

**Figure 2 materials-15-08279-f002:**
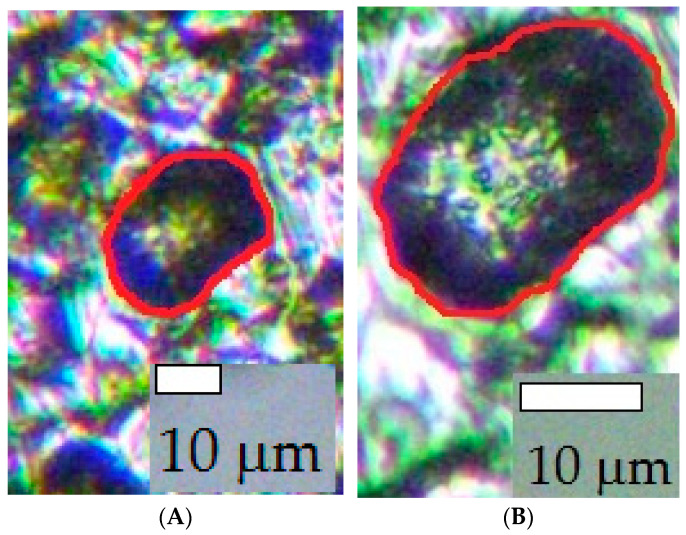
Images of the sample surface after SACP (*E*_a_ = −0.3 V, τ_a_ = 10 ms, *E*_c_ = −0.6 V, τ_c_ = 33 ms) in 3.5% NaCl solution. Magnification: 20× (**A**); 40× (**B**).

**Figure 3 materials-15-08279-f003:**
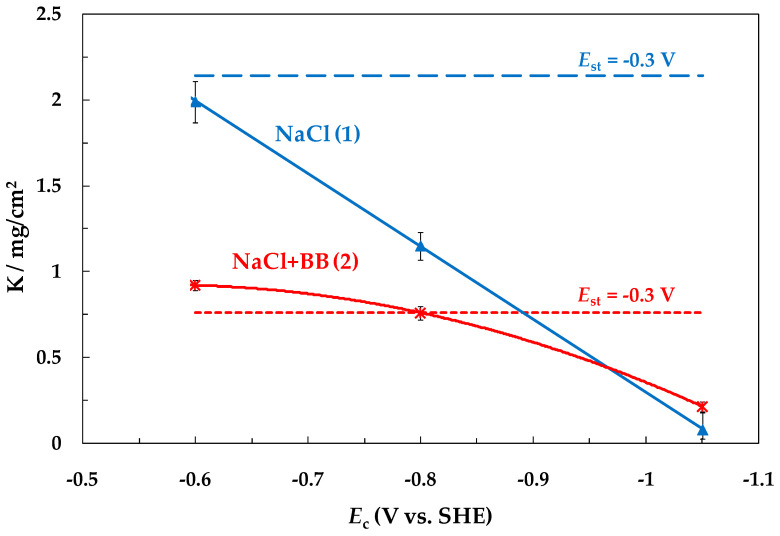
Dependence of steel weight loss on the SACP cathodic half-period potential (*E*_a_ = −0.3 V, τ_a_ = τ_c_ = 10 ms) in NaCl (1) and NaCl + BB (2) solutions. The setup consisted of 10^6^ SACP cycles. Dotted and dot-and-dash lines are explained in the text.

**Figure 4 materials-15-08279-f004:**
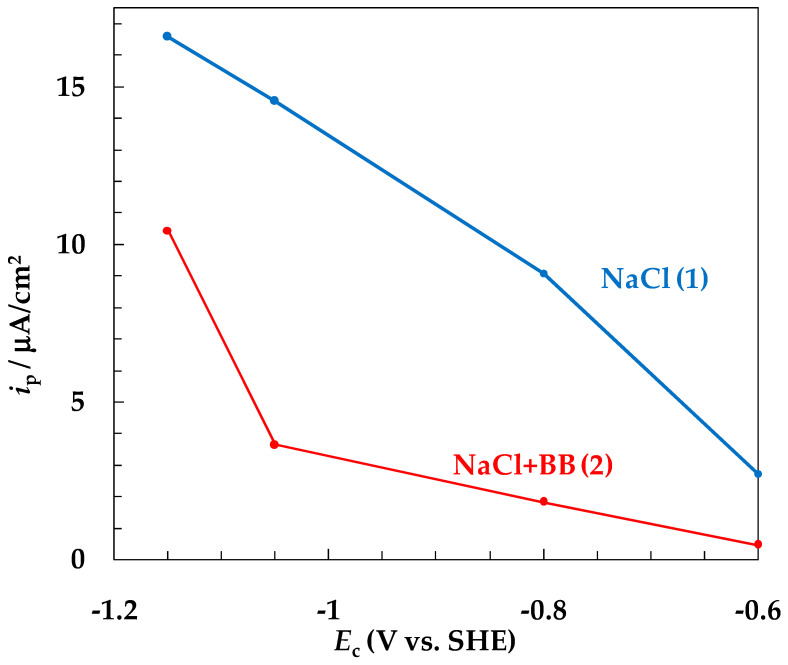
Variation in the rate of hydrogen penetration into steel depending on the potential of the SACP cathodic half-period (*E*_a_ = −0.3 V, τ_a_ = τ_c_ =10 ms) in NaCl (1) and NaCl + BB (2) solutions. The setup consists of 6 × 10^4^ SACP cycles.

**Figure 5 materials-15-08279-f005:**
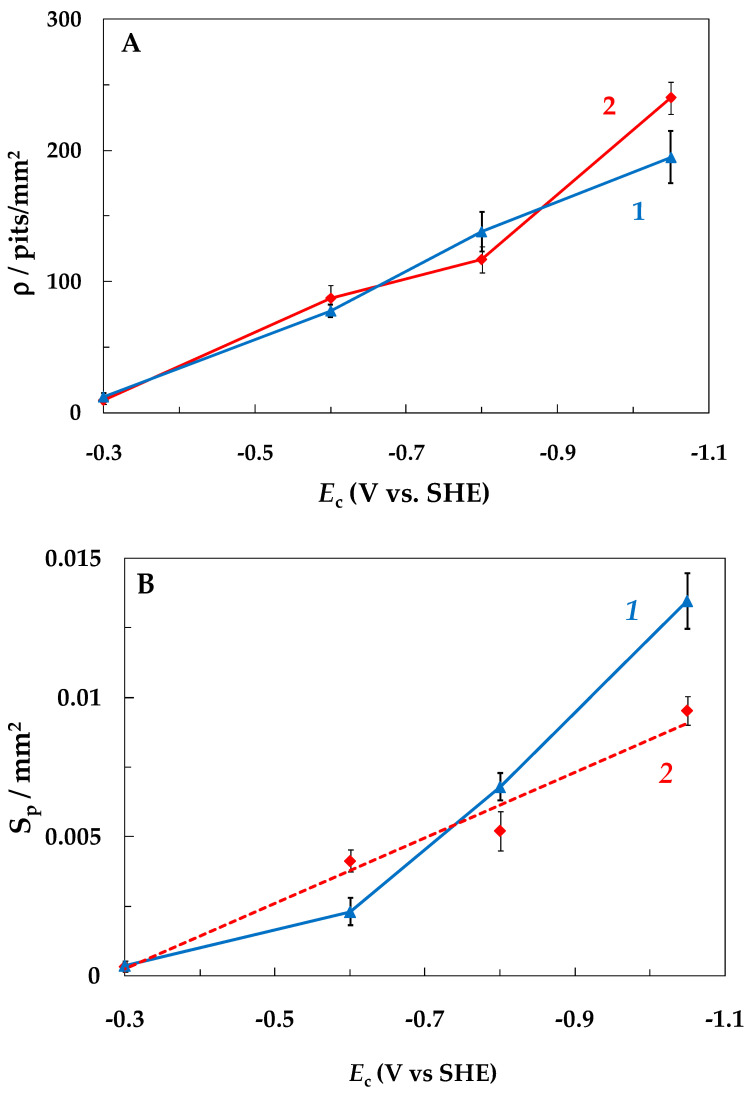
Variation in the density ρ (**A**) and total area *S_p_* (**B**) of the pits depending on the SACP *E*_c_ (*E*_a_ = −0.3 V, τ_a_ = τ_c_ = 10 ms) in NaCl (1) and NaCl + BB (2) solutions. The setup consists of 10^6^ SACP cycles.

**Figure 6 materials-15-08279-f006:**
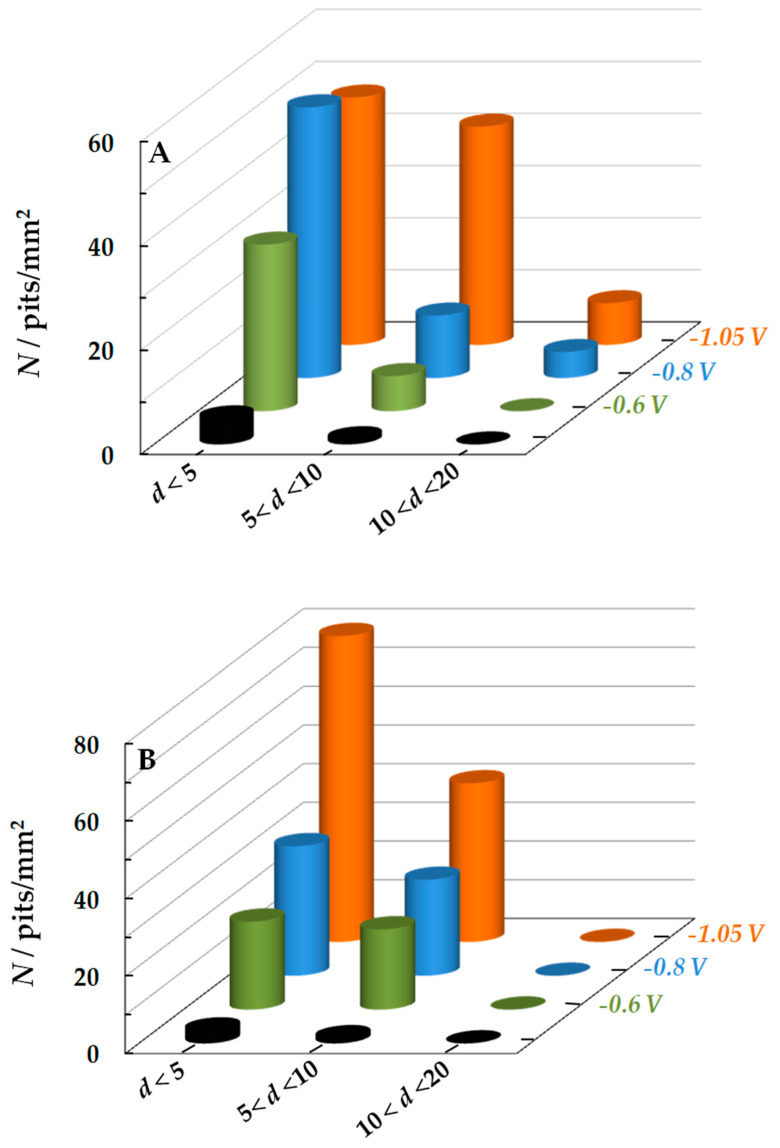
Number of pits *N* smaller than 5 µm, from 5 to 10 µm and larger than 10 µm in diameter *d* at various SACP *E*_c_ values in the NaCl (**A**) and NaCl + BB (**B**) solutions. *E*_a_ = −0.3 V, τ_a_ = τ_c_ = 10 ms. The number of pits at a constant anodic potential of −0.3 V is shown with black columns. The setup consisted of 10^6^ SACP cycles.

**Figure 7 materials-15-08279-f007:**
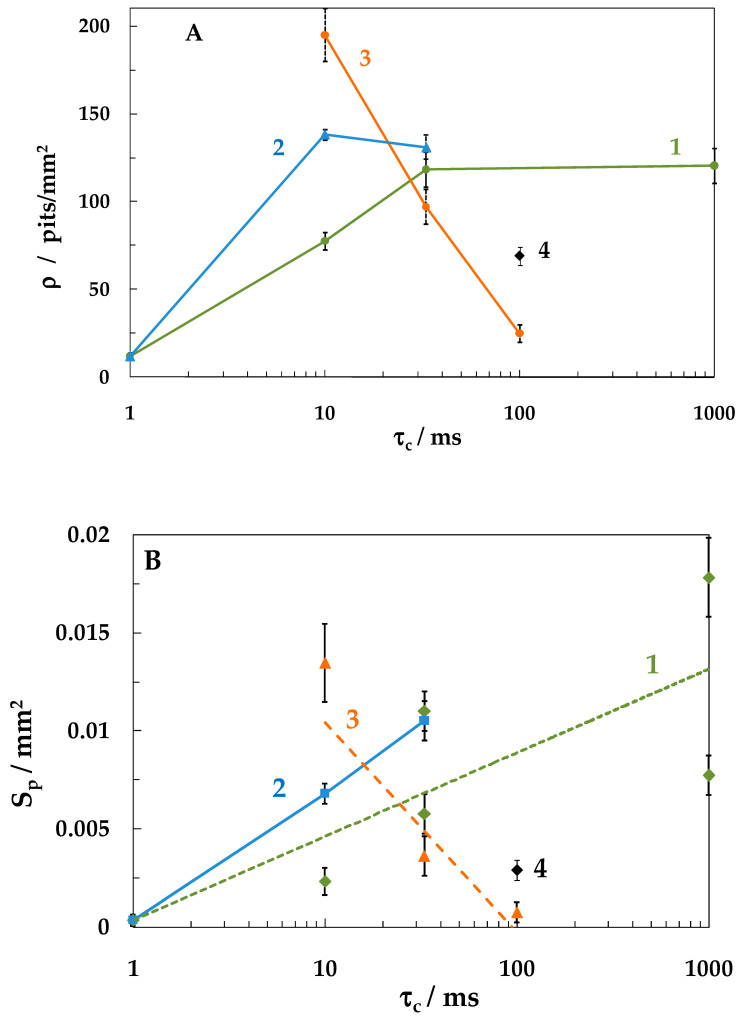
Variation in the density ρ (**A**) and total area *S*_p_ (**B**) of pits depending on the cathodic half-period duration τ_c_ at *E*_c_, V: −0.6 (1), −0.8 (2), −1.05 (3) and −1.15 (point 4). The setup consisted of 10^6^ SACP cycles at *E*_a_ = −0.3 V and τ_a_ = 10 ms in NaCl solution.

**Figure 8 materials-15-08279-f008:**
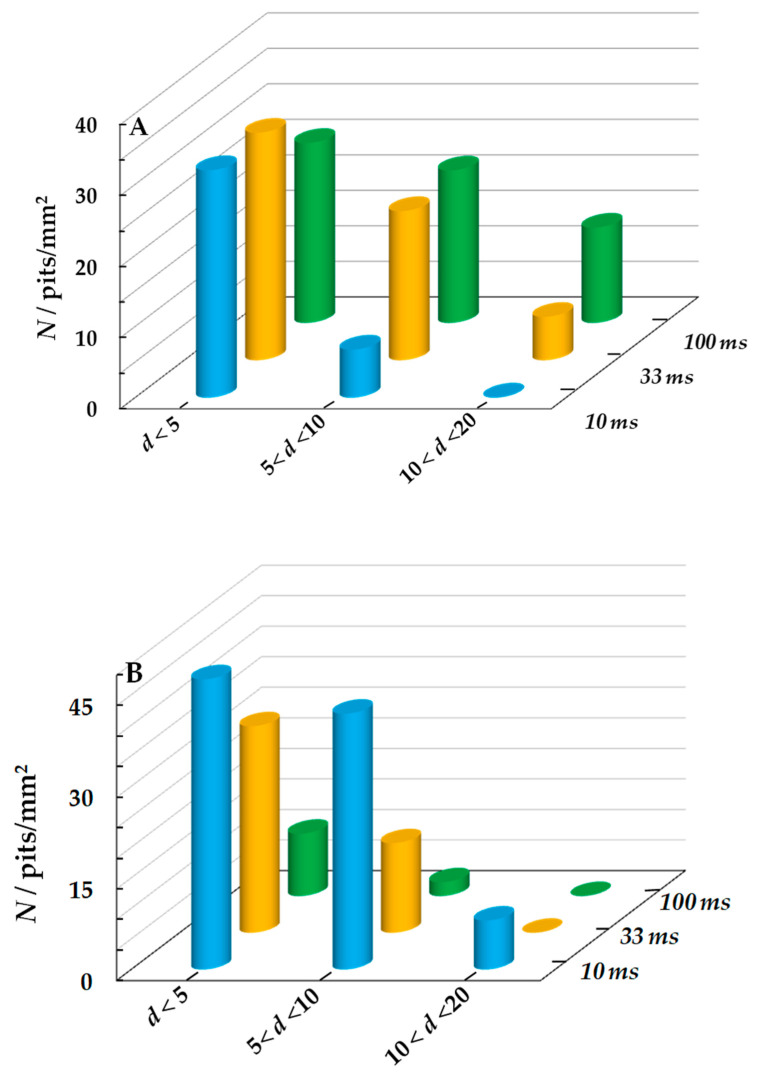
Number of pits *N* smaller than 5 µm, from 5 to 10 µm and larger than 10 µm in diameter *d* at various τ_c_ and *E*_c_ = −0.6 V (**A**) and −1.05 V (**B**). The setup consisted of 10^6^ SACP cycles with *E*_a_ = −0.3 V and τ_a_ = 10 ms in NaCl solution.

**Figure 9 materials-15-08279-f009:**
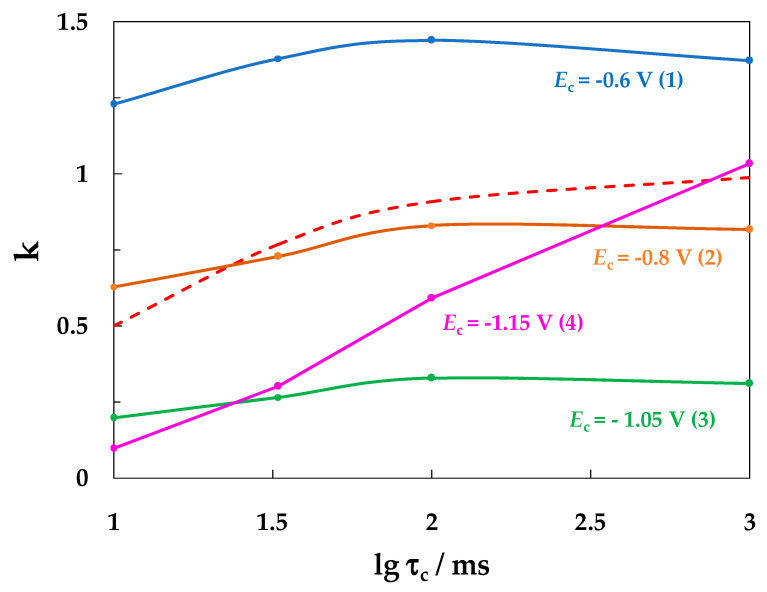
Variation in the *k* ratio vs. τ_c_ at *E*_c_, in V: −0.6 (1), −0.8 (2), −1.05 (3) and −1.15 (4). The setup consisted of 10^6^ SACP cycles at *E*_a_ = −0.3 V and τ_a_ = 10 ms in NaCl solution. The dashed line represents the curve calculated according to Equation (6); the explanations are provided in the text.

**Table 1 materials-15-08279-t001:** The chemical composition of 08 kp steel (wt. %).

C	Si	Mn	Ni	S	P	Cr	Cu	As	Fe
0.05	0.03	0.25	0.25	0.04	0.035	0.10	0.25	0.08	balance

## Data Availability

Not applicable.

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
