# Peer review of "Effect of Cathodic Protection Potential Fluctuations on the Corrosion of Low-Carbon Steels and Hydrogen Absorption by the Metal in Chloride Solutions with Nearly Neutral pH"

_materials, 2022, doi:10.3390/ma15238279_

Round 1
Reviewer 1 Report
This manuscript has great innovative significance in investigating Effect of cathodic protection potential fluctuations on the corrosion of low-carbon steels and hydrogen absorption by the metal in chloride solutions with nearly neutral pH. The work can arouse wide interests of researchers in design and preparation of new functional materials. The manuscript is interesting. In my frank opinion, the manuscript should be deserved for its final publication in such high-level Journal. The main reasons are as follows:
1. For ENGLISH ABSTRACT, the author should pay attention to the use of tense, and the manuscript can't use the simple present tense in the whole passage.
2. The research significance and future work should be described in the final stage of the abstract.
3. Aims need to be concisely stated and added at the end of introduction. Not only what was done/investigated, but why.
4. Authors have not mentioned borate buffer parameters in section 2. Please provide the data and also the corresponding reason for selecting such data? OR refer to the literature that has been reported?
5. Figure 1 is not clear, please provide a clear picture.
6. Please re-provide the ordinate of Figure 3 and 4.
7. In Fig. 5, If possible, please provide error bars.
Reviewer 2 Report
The work presented in this manuscript is dealing with important topic and can be published after minor correction.
1. Abstract need to revsied higlighting the new and achievements in numbers.
2. typos and grammars need to be addressed.
3. Authors have to enrich introduction section differnt approaches and materials reported anticorrosion.
4.Figures need to be replaced with high qulity ones.
Reviewer 3 Report
The manuscript entitled „Effect of cathodic protection potential fluctuations on the corrosion of low-carbon steels and hydrogen absorption by the metal in chloride solutions with nearly neutral pH“ is well structured and adequatilly written. The manuscript deals with efect of electric potential fluctuation caused most commonly stray currents on corrosion of low carbon steel. The results obtained from experimental research are well presented and explained in detail.
Although the manuscript is well written, the following corrections and clarifications are needed:
· At the end of the introductory section, the structure of the manuscript should be given.
· In the second section, it is necessary to provide the scheme of the exam circuit.
· Was the anodic half-period duration 10 ms for all cathodic half-period lengths (10, 33, 100 and 1000 ms)?
· Should Sp be written in equation (2) instead of S?
· What does Si represent in equation (3) and how is this parameter determined?
· It would be mathematically correct for the sum in equation (3) to be within certain limits.
· It is necessary to fix the mark on the ordinate of the diagrams shown in Figures 3, 4 and 6.
· I am unclear about the references in the first paragraph of the conclusion. Is this conclusion based on the experimental research presented in this manuscript or from the literature cited here? If the same conclusions were reached in the cited studies, this should be clearly indicated.
